# Cooperation in the face of thresholds, risk, and uncertainty: Experimental evidence in fisher communities from Colombia

**Juan C. Rocha**[1,2,3,4]*, **Caroline Schill**[1,2], **Lina M. Saavedra-Díaz**[5], **Rocío del Pilar Moreno**[6], **Jorge Higinio Maldonado**[6]

**1** Swedish Royal Academy of Sciences, Beijer Institute, Stockholm, Sweden, **2** Stockholm Resilience Centre, Stockholm University, Stockholm, Sweden, **3** Future Earth, Stockholm, Sweden, **4** South American Institute for Resilience and Sustainability Studies, Maldonado, Uruguay, **5** Programa de Biología, Universidad del Magdalena, Santa Marta, Colombia, **6** Facultad de Economía, Universidad de Los Andes, Bogotá, Cundinamarca, Colombia

* juan.rocha@su.se

**Data Availability Statement:** The experimental data necessary for replication is available in a public repository (10.6084/m9.figshare.12563549). Game data and survey data used in

## Abstract

Cooperation is thought to be a necessary condition to solve collective action dilemmas such as climate change or the sustainable use of common pool resources. Yet, it is poorly understood how situations pervaded by thresholds shape the behaviour of people facing collective dilemmas. Here we provide empirical evidence that resource users facing thresholds maintain on average cooperative behaviours in the sense of maximising their individual earnings while ensuring future group opportunities. A framed field experiment in the form of a dynamic game with 256 Colombian fishers helped us investigate individual behavioural responses to the existence of thresholds, risk and uncertainty. Thresholds made fishers extract less fish compared to situation without thresholds, but risk had a stronger effect on reducing individual fishing effort. Contrary to previous expectations, cooperation did not break down. If cooperation can be maintained in the face of thresholds, then communicating uncertainty is more policy-relevant than estimating precisely where tipping points lay in social-ecological systems.

## Introduction

Sustainability challenges are often characterised by situations pervaded by thresholds [1]. Achieving sustainable development goals such as eradicating poverty, dealing with climate change, and preventing the tragedy of the commons in using natural resources, require all cooperation to deal with situations characterised by non-linear dynamics with tipping points [2–5]. Under current development trajectories, ecosystems worldwide are at risk of undergoing more frequent and severe regime shifts –abrupt transitions in their function and structure– changing the flow of ecosystem services on which societies rely upon, and the source of livelihoods for many communities [6, 7]. Examples include bush encroachment, a regime shift that reduces the ability of ranchers to maintain cattle; soil salinisation which compromises the

this study has been anonymised. The code used for the analysis is publicly available at: https://github.com/juanrocha/BEST.

**Funding:** The research was supported by Formas (https://formas.se) grant 211-2013-1120 and 942-2015-731. The funders had no role in study design, data collection and analysis, decision to publish, or preparation of the manuscript.

**Competing interests:** The authors have declared that no competing interests exist.

ability of farmers to produce food; or the collapse of fisheries which could compromise the livelihoods of ∼ 51 million people who today depend on them, most of them from developing countries [8]. Over 30 different types of regime shifts have been documented in social-ecological systems, and their frequency and intensity are expected to increase [7, 9]. This raises the questions: how do people behave in situations pervaded by thresholds? How do thresholds affect individual decisions regarding the extraction from a shared resource? Do people race to the bottom and collapse their resources, or do they find strategies for dealing with threshold uncertainty?

Traditionally these questions have been studied from a rather theoretical point of view [4, 10–14] with a strong focus on public goods [5, 15–17]. Theoretical and empirical evidence suggests that the relationship between collective action and uncertainty is negative: the higher the uncertainty, the higher the likelihood of cooperation to break down [11, 12, 16, 18, 19]. Only under very specific circumstances in public good models, uncertainty was predicted to increase cooperation [5, 15]. However, most of these empirical results have been largely obtained in lab settings with "weird" subjects: western, educated, industrialised, rich, and democratic [17, 20, 21]. Whether these results hold when tested with people whose livelihoods depend on natural resources is still an open question.

To fill that gap, we designed a framed field experiment to investigate how resource users deal with different degrees of uncertainty regarding the existence of thresholds below which common pool resources can collapse [22]. Using group-level analysis focusing on sustainable resource use, we found that uncertainty around critical climate-induced thresholds is not necessarily bad but can in fact protect common pool resources [22]. The purpose of this paper is to test how individual resource users behave in situations pervaded by thresholds when facing collective action dilemmas. This allows us to investigate closely cooperative behaviour and the role of context for individual behaviour—both within the game and the fishers' everyday realities. This in turn allows us to test whether our previous results hold and identify critical factors and dynamics for managing common pool resources in situations pervaded by thresholds, risk and ambiguity.

## Methods

We played a dynamic common pool resource game with 256 fishers in 4 coastal communities of the Colombian Caribbean (see Appendix 1 in S1 File for instructions, and [22] for details about the communities). The game is inspired on previous lab experiments tested at Stockholm University [17, 21]. The game was framed as a fishery with the potential of a climate event to abruptly reduce the recovery rate of the fish stock on which the fishers' earnings depended.

At the time of the fieldwork and fieldwork preparation (2015/2016) there was not a formal ethical review process established at our institution. However, we made sure to follow relevant ethical requirements and practice for researchers in Sweden at the time e.g. as stipulated by the Act concerning the Ethical review of research involving humans (2003:460) and the Personal Data Act (1998:204). This means for example that we provided the participants with an informed consent containing information about the purpose of the study, how the data collected would be used and that the participants' anonymity would be guaranteed. We then only included participants that gave written consent (all did). We also only make an anonymised version of the game data and survey available, including only the variables used in this study for replication purposes.

### Fishing game

In the game, fishers made individual decisions each round of how much they wanted to fish from a common pool with 50 fish in the beginning. Communication was allowed and the social dilemma was faced in groups of four. The game lasted 16 rounds (unknown to the players), of which the last ten were run under one of three treatments or the control group (*baseline*). 64 fishers (16 groups) were randomly assigned to the *threshold* treatment, in which at the beginning of round seven a climate event occurred reducing the recovery rate of the fish below a stock size of 28 (threshold, Fig 1). For all other stock sizes the recovery rate remained the same. This framing is similar to hypoxia events –low water oxygen– which could follow times of drought or extreme rain, and have been recorded in the region for decades [23]. In times of hypoxia fish die creating death zones [24]. The second treatment was *risk*, where fishers (n = 64) knew that a climate event could occur in any following round reducing the fish stock's ability to reproduce with a 50% chance below the threshold. In the *uncertainty* treatment (n = 64) the same framing was used, but the probability of the climate event was between 0.1-0.9. The remaining fishers played the control condition (*baseline*, n = 64), which continued playing as in the first six rounds. The fish stock was restored to 50 fish at the beginning of round seven for all groups.

The climate event reduced the capacity of the fish stock to reproduce below a threshold. In the *baseline* the reproduction rate was 5 fish if the remaining fish stock was 5-19 or 35-45 fish, and 10 fish if the remaining fish stock was 20-34 (Fig 1). There was no reproduction in neither treatment for fish stocks below 5 or above 45, which was justified in the game as Allee effects. In too low densities, or highly populated ponds, fish have it harder to reproduce due to lack of partners or competition for resources. If the climate event occurred in the game, the reproduction rate changed to 1 fish for a remaining fish stock of 5-27, 10 fish for a remaining fish stock of 28-34, and 5 fish for a remaining fish stock of 35-45 (Fig 1). Once the climate event occurred, the reproduction rate changed for the rest of the game mimicking a long-lasting effect on the function and structure of the ecosystem—a regime shift.

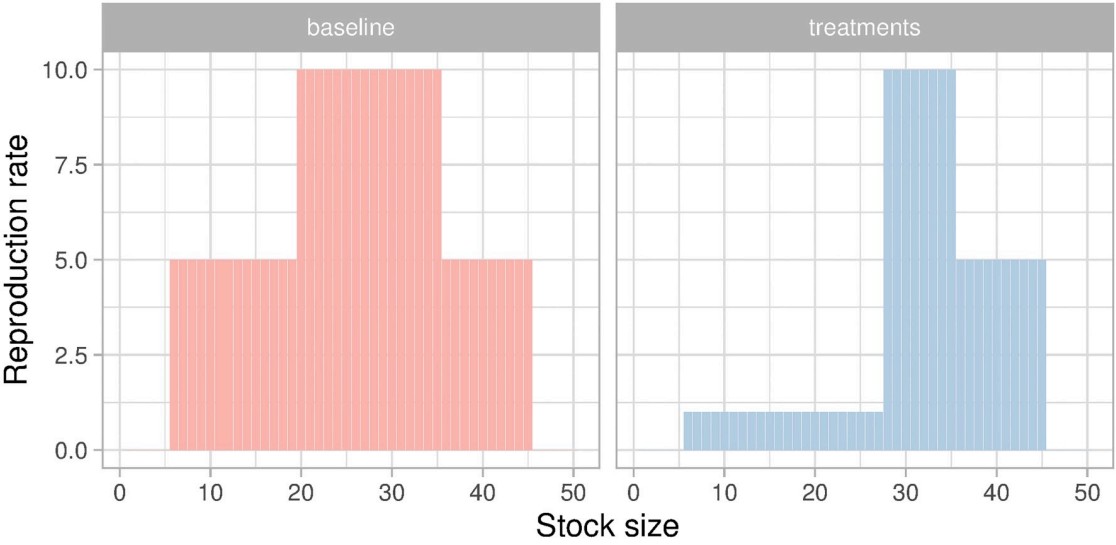

**Fig 1. Reproduction rate as function of stock size.** Treatments where the climate event triggers the threshold effect (threshold, risk, and uncertainty) can have a lower reproduction rate if the stock size is below 28 fish.

We communicated *risk* and *uncertainty* with a ballot system to avoid deception. For *risk*, five green and five red stones were shown at the beginning of the round. We drew one stone in private. If it was red the climate event occurred and we calculated the reproduction rate at the end of the round accordingly. If the stone was green, we kept the reproduction scheme of the *baseline* (Fig 1). Since we drew the stone in private, fishers could not know if the climate event happened if the remaining stock was above the threshold $\theta = 28$ since both reproduction schemes are identical for $S_t \geq 28$. For the *uncertainty* treatment, we showed them ten red and ten green stones. We first took one stone of each colour and put them into an urn. The remaining 18 stones were mixed in another urn. Once mixed, 8 stones were moved to the first urn without revealing their colour, so neither experimenters or fishers knew the exact distribution of stones of the urn we later used to draw the climate event. All we knew was that the probability could be between 0.1 and 0.9 since for sure there was one green and one red stone in the urn. For the *risk* and *uncertainty* treatments, we drew a stone every round regardless if the climate event occurred or not, and the stone was returned to the urn so each round had exactly the same odds.

The fishing game was part of a 3 hour workshop that was carried out in four Colombian fishing communities in the Caribbean coast in February 2016. Participants knew that the total duration of the workshop was 3 hours but they did not know how long the fishing game would last. This was to avoid so-called end of game effects—people depleting the resource to maximise their individual earnings. Each workshop consisted of the fishing game (including two to three practice rounds), a post-experimental survey, and a risk/ambiguity elicitation task. Before starting, each participant signed a consent form agreeing to participate in all three activities and authorising us to use the anonymised data for research purposes.

To make decisions in the game more realistic, each fisher earned $COL500 (USD$0.14) for each fish caught, in addition to a show-up fee (COL$15000, USD$4.3) meant to compensate for the time invested in the workshop. A day spent in the workshop meant for them a day without fishing, so their average earnings were adjusted in a way that represented a typical working wage. The full instructions of the game (English version) are available in Appendix 1 in S1 File (Appendix 3 in Spanish of S2 File). For a more detailed explanation of the experiment and how it relates to similar experimental designs, please see [22]. Our selection of sample in terms of the number of groups per treatment (64) was informed by a previous study with a similar experimental design performed in the lab with students at Stockholm University [17, 21]. The within and between group variances, however, were higher in the field, resulting on a slightly lower power in our field data (0.74 versus 0.8).

## Surveys

After the game, each fisher participated in a 56-question survey. Such post-experimental surveys commonly complement behavioural economic experiments both in the lab and in the field [25]. Our survey is inspired by surveys we, the author team, used for previous studies [17, 19] but adapted to our research question. The purpose of the survey was to better investigate the context in which individual decisions are taken—both with respect to the fishers' perceptions about the game and its dynamics as well as to the fishers' everyday reality. The survey was divided into five sections. The first section was about the game and their perceptions about the activity, for example, whether they expected the game to end when it did. The second section was about their fishing habits: how much effort they put on fishing (time per day or year), how much earnings they get in a good or bad day, whether they own and share the fishing gear, whether they fish in groups, or what are the targeted species. The third section was

about traditional ecological knowledge focused on questions about abrupt changes in their fishing grounds in the past and the type of strategies they have used to cope with it. The fourth section was about cooperative activities and associations in the community. The last section included questions about demographics, socioeconomic information, and sense of place. The full questionnaire is available in Appendix 2 in S1 File.

### Risk and ambiguity elicitation task

After the survey fishers were asked to do a risk and ambiguity elicitation task [26]. To measure risk and ambiguity aversion we asked fishers to choose two times between six binary lotteries: $13000|$13000; $10000|$19000; $7000|$25000; $4000|$31000; and $0|$38000. For risk, the chances of getting the high payoff was 0.5, while for ambiguity it was an unknown probability between 0.1-0.9. Half of the fishers started with the risk task and the other half with the ambiguity task in order to control for order effects. Their choices were transformed to a discrete variable used in our regressions that takes one if the fisher is risk or ambiguity averse (when the $13000|$13000 lottery was chosen), and six when the fisher is risk or ambiguity keen (when the $0|$38000 lottery was chosen). The payoffs from the risk and ambiguity elicitation task were paid to only one fisher per group (decided by lottery).

### Regressions

We fitted a random effects panel model to our full game dataset (N = 4096) to disentangle treatment effects with a difference-in-difference regression. It follows the form:

$$Y_{i,t,g} = \mu_{i,t,g} + \gamma G_{i,t,g} + \delta T_{i,t,g} + \tau G_{i,t,g} T_{i,t,g} + \epsilon_i + \epsilon_t + \epsilon_g \tag{1}$$

where $\gamma$ is the effect of being assigned to a group with a treatment, $\delta$ is the effect of the treatment (before-after), and $\tau$ is the interaction term that captures the average treatment effect on the treated. As response variables $Y_{i,t,g}$ we used individual extraction, proportion of stock extracted, and cooperation (defined below). The average treatment effect on the treated (ATT, Fig 2) in the difference-in-difference framework was calculated according to the following definitions (Table 1):

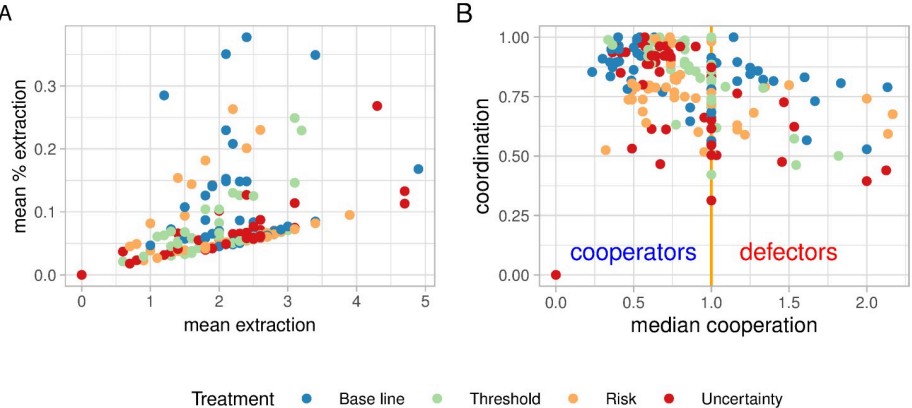

**Fig 2. Fishers fish less and cooperation does not break down.** Effects of treatments (threshold, risk, uncertainty) on individual extraction (top), proportion of stock (middle) and cooperation (bottom) compared to baseline (no potential for thresholds). Treatment effects are tested with a difference-in-difference random effects model with (respect to individual extraction, the proportion of the stock extracted, and cooperation (N = 4096) per response variable. Before refers to rounds before the introduction of the treatments (round 1-6) and after refers to the rounds after introduction of the treatments (round 7-16). The counterfactual is the expected response of fishers in the treatment if they would have played the baseline instead. S1-S3 Tables in S1 File complement this figure with a sensitivity analysis of robust standard error estimations.

**Table 1. Difference-in-difference specifications.**

| Terms | After ($T_i = 1$) | Before ($T_i = 0$) | After-Before |
|---|---|---|---|
| Treated $G_i = 1$ | $\hat{\mu} + \hat{\gamma} + \hat{\delta} + \hat{\tau}$ | $\hat{\mu} + \hat{\gamma}$ | $\hat{\delta} + \hat{\tau}$ |
| Control $G_i = 0$ | $\hat{\mu} + \hat{\delta}$ | $\hat{\mu}$ | $\hat{\delta}$ |
| Treated-Control | $\hat{\gamma} + \hat{\tau}$ | $\hat{\gamma}$ | $\hat{\tau}$ |

## Cooperation: Individual behaviour in context

To gain a better understanding of the interplay between group-level dynamics, and the context in which each individual decision was made [27], we designed two additional response variables: cooperation and coordination. Broadly speaking cooperation is working together towards a shared goal. Cooperation can also be defined as *"a form of working together in which one individual pays a cost (in terms of fitness, whether genetic or cultural) and another gains a benefit as a result"* [28]. In the context of common pool dilemmas (and non-dyadic games) cooperation can be interpreted as favouring the common good over individual benefits [29, 30]. An important distinction in the literature is that of cooperators versus defectors, while cooperators pay a cost for other(s) to benefit, defectors have no cost and do not deal out benefits [31, 32]. Here we operationalise these definitions by measuring cooperation as the ratio of the individual extraction $x_{i,t}$ with respect to the optimal level for the group. Thus, cooperation happens when (i) individuals take a number of fishes that maintain the fish stock at the optimal level (i.e. above the threshold in the treatments), or when (ii) take no fishes when the fish stock is below the optimal level (i.e. below the threshold in the treatments). Cooperation $C$ is measured assuming fairness or equal sharing of the stock available for fishing $S_t$ (above $\theta = 28$ in treatments and $\theta = 20$ in *baseline*):

$$C_{i,t} = \frac{x_{i,t}}{\frac{S_t - \theta}{N}} \tag{2}$$

where $N$ is the number of players in the group (always four in our experimental design). To avoid division by zero or negative values, when the fish stock is below the optimal level (20 or 28 depending on treatment, denominator < 1) and fishers do not take any fish ($x_{i,t} = 0$), we consider that they cooperate $C = 1$ (212/4096 observations), and if the denominator is zero and they take one fish ($x_{i,t} = 1$) cooperation is set $C = 1.5$ (17/4096 observations). Thus, cooperation is at its maximum when $C = 1$ meaning that the individual took 100% of what was fair to take while maintaining the optimal stock level and avoiding crossing the threshold in the treatments. If cooperation $C < 1$ the fisher did cooperate in order to avoid the threshold but was not efficient at maximising her/his personal utility; if $C > 1$ the fisher did not cooperate and preferred maximising her/his utility over the common good in the long run. If $C = 2$ the individual took twice as much as it was fair to take, and by doing so the group could have crossed the threshold. Cooperation in this interpretation is not understood as a point in time but by the distribution it forms over time. However, in any given round, the value of cooperation can be > 1 because a fisher can take one or two extra fish by agreement (e.g. a rotation scheme), by having weak agreements that do not specify quotas (e.g. "let's fish less"), or by mistake. Crossing the threshold is, however, the aggregated effect of individual decisions. For that reason, we also introduced coordination as the average (Bray-Curtis) similarity distance to other group members' decisions through the game. Thus, if coordination is close to one the individual extraction $x_{i,t}$ is very similar to the other group

members, while if coordination is close to zero, $x_{i,t}$ is very dissimilar to the rest of the group (S1 Fig in S1 File).

## Results

### Fishers fish less and cooperation does not break down

Fishers facing thresholds tend to fish less compared to the baseline both in absolute terms as well as in proportion to the availability of the resource. We also find that contrary to theoretical expectations, cooperation does not break down. We studied the individual behaviour of fishers by looking at their individual extraction $x_{i,t}$, the proportion of the stock they appropriated per round ($x_{i,t}/S_t$), and their levels of cooperation $C$. A difference-in-difference panel model with random effects reveals that treatment effects for individual extraction and proportion of stock extracted are significant and negative (Fig 2). The model also shows that treatment effects on cooperation are not significant. A Hausman test suggests that our choice for random effects is preferred for the proportion of stock available and cooperation ($p > 0.05$), but it supports fixed effects for individual extraction ($p < 0.05$). However, since our panel is nested, we fitted a random-effects model clustered around individuals, groups, and time following our hierarchical design. A fixed-effects model would have not allowed us to control for the different levels of nestedness. A Breusch-Pagan Lagrange multiplier test further supported our choice of a random model when compared with a pooled regression with any of the response variables ($p << 0.05$).

The reduction of fishing effort is stronger for *risk* than for *threshold* or *uncertainty* treatments. Our results are robust to different choices of clustering standard errors (see S1-S3 Tables in S1 File) which were clustered simultaneously around individuals, groups and time. Given the nested structure of our design and that decisions in the past affect the stock size in the future, we expected that our dynamic game data presented cross-sectional dependence. We confirmed these expectations with a Breusch-Pagan LM test for cross-sectional dependence ($p << 0.05$ for all response variables) and a Breusch-Godfrey/Wooldridge test for serial correlation ($p << 0.05$ for all response variables). In addition, a Breusch-Pagan test reveals that our models are heteroskedastic ($p < 0.05$), meaning that the variances change over time. To correct for heteroskedasticity, cross-sectional dependence, and serial correlation, we calculated robust standard errors by estimating the variance-covariance matrix with heteroskedasticity and autocorrelation consistent estimators (S1-S3 Tables in S1 File). We also performed a F-test to the joint linear hypothesis $H_0$: $\gamma + \tau = 0$, this is that the difference in the coefficients before and after treatments (*threshold*, *risk*, and *uncertainty*) are indeed different from zero. We found that our differences are significant for individual extraction ($F = 5.95$, $p << 0.05$, $df = 3$), weakly significant for proportion of stock extracted ($F = 2.23$, $p = 0.08$), and non-significant for cooperation ($F = 0.3$, $p = 0.8$). When tested individually for each treatment in the case of proportion of stock extracted, the weakly significant treatment was *risk* ($p = 0.08$), while *threshold* and *uncertainty* were both significant ($p = 0.02$, $0.01$ respectively; Fig 2).

Besides the effects of treatments on the reduction of fishing effort, we find that cooperation does not break down (Fig 2). While these results already contradict the premise that uncertainty breaks down cooperation, our response variables thus far do not allow us to investigate the context in which each decision was taken. For example, agreements or the emergence of rules are ignored, and an amount of fish caught worth the same in the above regression if they are caught before or after crossing potential thresholds. In the game and real life they are not the same thing. The same amount of fish extracted can have substantially

different impacts on the stock size and the potential earnings of fishers if non-linear thresholds are crossed (Fig 1).

## Fishing decisions in context

To better understand what explains the behaviour of individuals in terms of cooperation and coordination, we regressed variables that summarize individual behaviour from the second part of the game against explanatory variables that were individual attributes. As dependent variables we used median cooperation, coordination, the mean extraction, the mean proportion of the stock extracted, and their variances (Fig 3, see S1 Fig in S1 File for correlations between response variables). Decrease in variances and increase in coordination can be seen as empirical proxies of the emergence and compliance of agreements. As explanatory variables, we used our treatments, after controlling for socioeconomic variables (e.g. education, income), risk and ambiguity aversion (See Methods), the percentage of rounds in which individuals made agreements (a proxy of the intention but not necessarily of agreement compliance), and place to account for fixed effects that were not necessarily controlled for with our socioeconomic terms. Since our experimental design focuses on the impacts of tipping points in natural resource dynamics, we approximated income not as the amount of money people make per month, but rather as the frequency of bad days (i.e. returning from a fishing trip without any earnings). The latter although collinear with reported income, is a better proxy of exposure to regime shifts. We also include a response variable about the expectation of the fisher's children to depend on fishing as livelihood to deal with the long term perspective of sustaining the

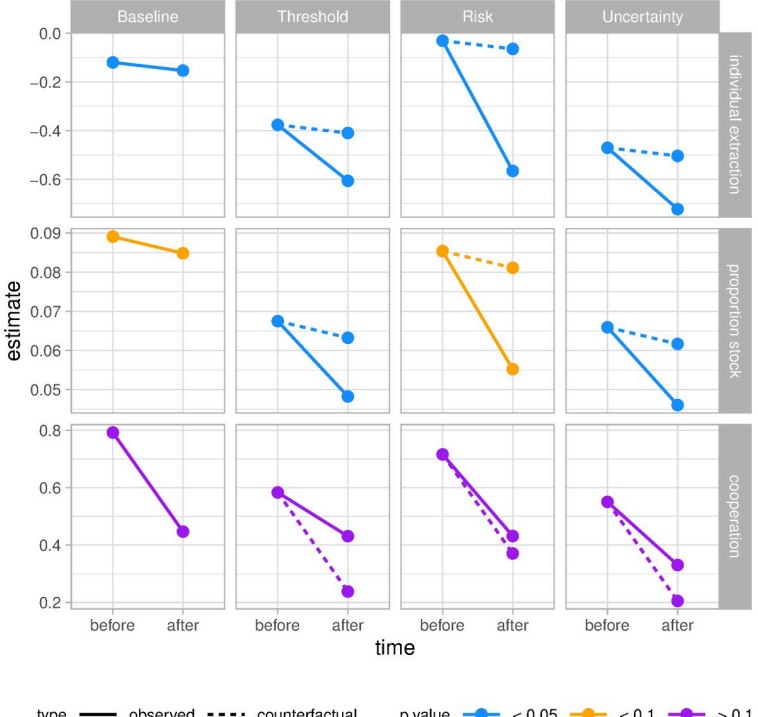

**Fig 3. Relationships between response variables of individual behaviour.** Figure A) shows the relationship between cooperation and coordination, figure B) shows the relationship of mean extraction and the mean proportion of the extraction. Each point represent an individual player (N = 256) and the summary statistic calculated over the second part of the game (rounds 7-16, N = 2560).

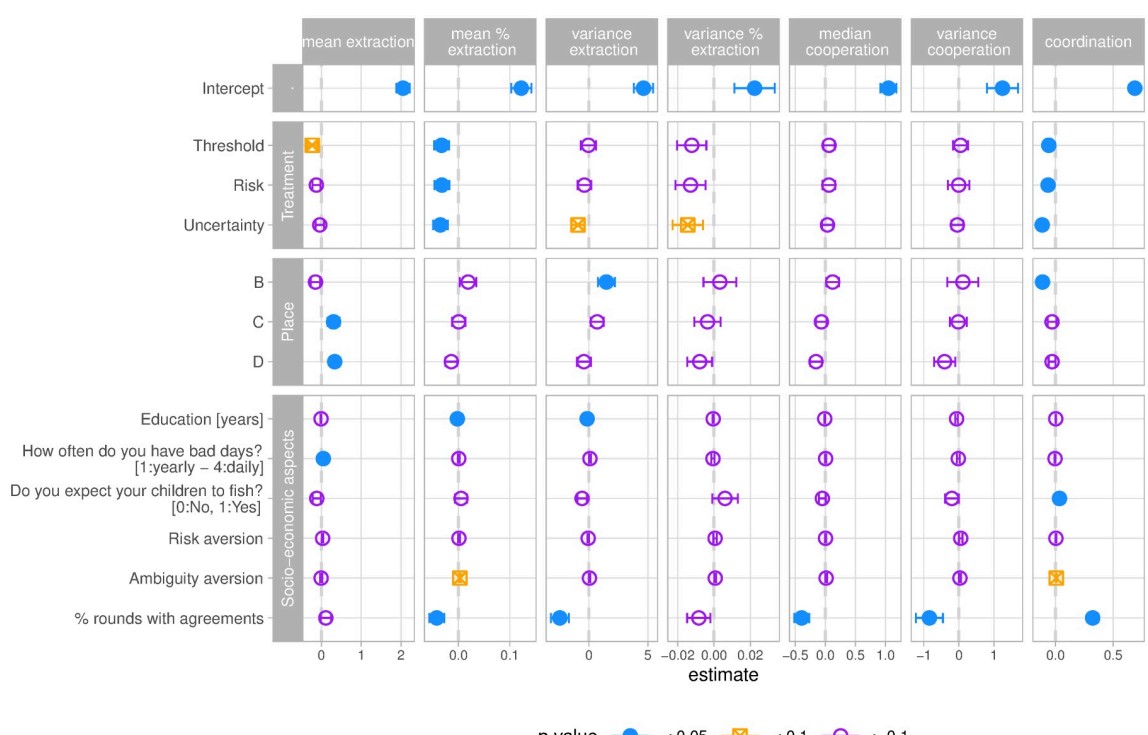

**Fig 4. Individual behaviour as function of treatments and different contextual factors.** The panel summarises results from an OLS regression for each of the response variables. Treatment effects are shown by taking into account place (A-D), group dynamics (e.g. reaching agreements), and several socioeconomic aspects. S4 Table in S1 File complement this figure with precise estimates and summary statistics. Error bars denote 95% confidence intervals calculated with a CR2 robust standard errors estimator.

resource, as well as group fishing and sharing of fishing arts to control for aspects of the fishing activity that can prime individuals to be more cooperative (shown in S4 and S5 Tables in S1 File).

We find that all treatments significantly reduced the proportion of stock that fishers extracted (Fig 4). Individuals who played the uncertainty treatment decreased coordination, yet coordination increased in groups that reached agreements. The proportion of rounds with agreements (intentions) had a negative effect on the proportion of stock extracted, the variance of extraction, and the median and variance of cooperation suggesting that agreements were in average followed. Fishers who reached agreements were better at maximising their individual earnings while maintaining sustainable stock levels (Fig 4). Median cooperation and its variance were only affected by the proportion of rounds people reached agreements, showing that it responds more to in-group dynamics rather than treatments or socioeconomic effects. We also found place effects that were not accounted for by our socioeconomic controls, showing that place B had on average less coordination and higher variance of extraction, while place D had higher extraction and higher cooperation ($C \leq 1$), both when compared to place A. People with higher levels of education reduced their variance of extraction, while people with a higher frequency of zero income days tend to fish more, but these effects are relatively small. Controlling for fishing art sharing, risk or ambiguity aversion render weakly significant coefficients ($p < 0.1$) and their effect sizes are relatively small together with other socioeconomic controls (S4 Table in S1 File). Controlling for individual behaviour in the first part of the game is significant for most of our response variables (except variances S4 Table in S1 File), suggesting that individuals bring cooperative preferences to the game that are independent of our treatments

and other socio-economic factors. Some of our socio-economic factors are partially correlated with place (S1 Fig in S1 File), thus S5 and S6 Tables in S1 File reproduce the regression without place and only place terms respectively.

## Discussion

Fishers under uncertain thresholds showed lower levels of extraction than when the threshold was known. Risk had a stronger effect at reducing individual fishing effort than uncertainty. Cooperation was not affected by thresholds, risk or uncertainty. This result supports and complements previous findings that uncertainty around critical climate-induced thresholds is not necessarily bad but can in fact protect common pool resources [22], and expand our understanding of individual-level dynamics that were not accessible in group-level studies [22, 33]. Our central result contradicts previous theoretical and empirical findings that predicted break down of cooperation under situations with threshold uncertainty [11, 12, 16, 19]. Our findings support the hypothesis that uncertainty can increase cooperative behaviour in public goods settings when the value of the public good is sufficiently high [5, 15], by means of reducing exploitation effort. Our experiment is not a public good setting, but it can be translated to a common pool resource when the dependency of the resource is sufficiently high. Previous work has concentrated their efforts on settings with western, educated, industrialized, rich and democratic individuals [20]. Here we empirically show that the negative relationship between cooperation and uncertainty does not hold for common pool resource games, played with resource users whose livelihoods largely depend on natural resources. On the contrary, our study supports a small but growing body of empirical evidence suggesting that uncertainty can help protect the commons when ecosystems are susceptible to thresholds such as climate-induced regime shifts [22, 33].

One potential explanation for the deviation from theoretical expectations can be personality traits [34, 35]. We expected that risk and ambiguity aversion could be key personal traits affecting individual behaviour. Our results suggest however that group dynamics seem to override personal preferences regarding risk and ambiguity aversion. Some resource users tend to have pro-social and pro-environmental behaviour, others have more individualistic or short term preferences (Fig 3); but as observed by a previous study in the same region, pro-social fishers are less likely of changing their behaviour than non-cooperators [19]. This in turn scales up to the group level, where groups with higher proportions of cooperative individuals maintain higher levels of fish stock despite an occasional free-rider [19]. Our results suggest that fishers were responding more to in-group dynamics (e.g. increasing coordination) and personal preferences regarding pro-social behavior, rather than risk or ambiguity aversion.

Our study shows that reaching agreements decreases fishing efforts and increases cooperation. It suggests that a common strategy that evolved in the game was approaching the threshold without crossing it, thus maximising both social and individual benefits. By reducing fishing effort or keeping close to the social optimal people do cooperate. However, cooperation–as measured in our study–was not affected by our treatments. Cooperative behaviour then seems to be driven more by personal preferences and group dynamics than levels of uncertainty. This observation agrees with previous experiments studying internal Nash solutions on common pool resources [19], and highlights the important and well established role of communication in providing groups an arena for agreement negotiations, rule making, social pressure, and coordinating actions [17, 22, 30, 36]. Previous participatory research in the communities studied supports with different methods our findings [37, 38]

Fishers do reduce fishing in presence of thresholds, but the effect occurs to a lesser extent when uncertainty is high. As shown in [22], this is partly due to our experimental design

where uncertainty can mask free-riding behaviour and slow down the erosion of trust. In that sense, the uncertainty about thresholds also induces social uncertainty about adhering to agreements. An alternative explanation is that under higher levels of uncertainty fishers adopt a more exploratory mode (higher variance) with less strict agreements. Reduced variance of decisions over time and increased coordination across group members suggest that people with strong agreements (e.g. strict quotas) were more successful on maintaining the stock above the threshold than groups with soft agreements (e.g. "let's fish less"). Further research efforts could target disentangling the effects of the different forms of uncertainty regarding the dynamics of the natural resources with potential thresholds, the social uncertainty about free-riding, or the effects of norms ambiguity. As this type of experiments scale up to more realistic settings, noise induced by social network structures needs to be taken into consideration realising that humans have limits to social interactions [39], and that social relationships are heterogeneous in number and quality.

## Conclusion

If the existence of thresholds already triggers pro-environmental behaviour reducing fishing effort in natural resource users, then communicating their potential effects on ecosystems and society is more important than quantifying the precise point at which ecosystems tip over. Tipping points are difficult to observe and quantify in nature, they are not unique and they are expected to interact with other tipping points [7, 40, 41], meaning that their exact points change over time. While precise measurements can be out of reach specially in settings where monitoring programs are weak or not in place (e.g. developing countries), knowledge about the circumstances under which an ecosystem can tip over can already trigger behavioural change for maintaining natural resources in configuration that provide crucial ecosystem services for livelihoods. In our case study, these circumstances are related with high concentrations of nutrients in water often correlated with use of fertilizers in agricultural activities, or periods of high sediment input following droughts and strong rainy seasons such as ENSO events [23, 24, 42]. Identifying such circumstances and communicating uncertain but potential regime shifts can mobilise social action towards sustainable behaviour in natural resource users.

## Supporting information

**S1 File.**
(PDF)

**S2 File.**
(PDF)

## Acknowledgments

We would like to thank the fishing communities that participated in our experiments. This work received valuable feedback on early stages of its design by Anne-Sophie Crépin, Therese Lindahl, Juan Camilo Cárdenas, Maria Alejandra Velez, and Sandra Vilardy. The field work would have not been possible without the support of Nidia Vanegas, Alisson Soche, Darlin Botto, María de los Ángeles González, Cristhian Marrugo, Gloria de León, Jaime González, and Jesús Jiménez. JCR benefit from technical advise by Joshua Abbott. The research was supported by Formas grant 211-2013-1120 and 942-2015-731.

## Author Contributions

**Conceptualization:** Juan C. Rocha, Caroline Schill, Rocío del Pilar Moreno, Jorge Higinio Maldonado.

**Data curation:** Juan C. Rocha.

**Formal analysis:** Juan C. Rocha.

**Investigation:** Juan C. Rocha, Caroline Schill, Lina M. Saavedra-Díaz.

**Methodology:** Juan C. Rocha, Caroline Schill, Lina M. Saavedra-Díaz, Rocío del Pilar Moreno, Jorge Higinio Maldonado.

**Project administration:** Juan C. Rocha, Caroline Schill.

**Resources:** Lina M. Saavedra-Díaz.

**Software:** Juan C. Rocha.

**Visualization:** Juan C. Rocha.

**Writing – original draft:** Juan C. Rocha.

**Writing – review & editing:** Juan C. Rocha, Caroline Schill, Lina M. Saavedra-Díaz, Rocío del Pilar Moreno, Jorge Higinio Maldonado.

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
