## [Decision Letter · Decision Letter 0]

20 Aug 2020

PONE-D-20-19709

Cooperation in the face of thresholds, risk, and uncertainty

PLOS ONE

Dear Dr. Rocha,

Thank you for submitting your manuscript to PLOS ONE. After careful consideration, we feel that it has merit but does not fully meet PLOS ONE’s publication criteria as it currently stands. Therefore, we invite you to submit a revised version of the manuscript that addresses the points raised during the review process.

ACADEMIC EDITOR: The two reviewers have provided constructive and detailed comments. They both agreed that the work is intersting, relevant and would provide a strong contribution in terms of real-world evidence to support theoretical findings in behavioural modelling research. However, there are several aspects of the paper that need improvements, for which the reviewers have provided constructive suggestions. Please carefully consider them in the revision of your manuscript.

We look forward to receiving your revised manuscript.

Kind regards,

The Anh Han, Ph.D.

Academic Editor

PLOS ONE

Journal Requirements:

5. Please amend your list of authors on the manuscript to ensure that each author is linked to an affiliation. Authors’ affiliations should reflect the institution where the work was done (if authors moved subsequently, you can also list the new affiliation stating “current affiliation:….” as necessary).

6. We noted in your submission details that a portion of your manuscript may have been presented or published elsewhere. [We are also submitting another manuscript based on the same data entitled “Uncertainty can help protect the local commons in the face of climate change” also available as a preprint at https://papers.ssrn.com/sol3/papers.cfm?abstract_id=3468677. ] Please clarify whether this [conference proceeding or publication] was peer-reviewed and formally published. If this work was previously peer-reviewed and published, in the cover letter please provide the reason that this work does not constitute dual publication and should be included in the current manuscript.

Additional Editor Comments (if provided):

The two reviewers have provided constructive and detailed comments. They both agreed that the work is intersting, relevant and would provide a strong contribution in terms of real-world evidence to support theoretical findings in behavioural modelling research. However, there are several aspects of the paper that need improvements, for which the reviewers have provided constructive suggestions.

Reviewers' comments:

Reviewer's Responses to Questions

**Comments to the Author**

1. Is the manuscript technically sound, and do the data support the conclusions?

Reviewer #1: Partly

Reviewer #2: Yes

2. Has the statistical analysis been performed appropriately and rigorously? 

Reviewer #1: Yes

Reviewer #2: Yes

3. Have the authors made all data underlying the findings in their manuscript fully available?

Reviewer #1: Yes

Reviewer #2: Yes

4. Is the manuscript presented in an intelligible fashion and written in standard English?

Reviewer #1: Yes

Reviewer #2: Yes

5. Review Comments to the Author

Reviewer #1: The authors present and discuss empirical results of an experiment with fisherman in Columbia, conducted to study individuals’ decision in a common-pool resource dilemma. Particularly, the authors design treatments to better understand how cooperation depends on shocks in the resource growth, thresholds that determine that resource growth, uncertainty and risk. The authors show that, in all the conditions tested, fisherman cooperate more after the resource growth regime shifts. Notwithstanding, the risk condition is the one presenting a higher increase in cooperation. The authors conclude that uncertain thresholds increase the level of cooperation.

This work has several positive aspects: 1) first of all, it is noteworthy that the common-pool resource game is actually played by individuals that deal with such dilemmas in their daily lives; this makes, in my opinion, the collected data extremely valuable. 2) second, it is also remarkable that the experiments were conducted with “non-weird” subjects. 3) last but not least, the results are interesting in that they point out that uncertain thresholds increase cooperation. The analysis performed appears to be sound and the results are, as far as I know, original.

Despite several worthy features, I also found some unclear points in the manuscript:

1) to start with, the authors use a very specific growth structure, characterized by (not one, but) four thresholds defining different fish stock growth rates. What was the reason behind such specific thresholds and growth rates?

2) related with the previous point: what are the general characteristics of the interaction that results from such thresholds, growth rates and number of players per group? (e.g., what would be a pareto optimal set of strategies? or fair? or Nash equilibria?). The authors provide metrics of cooperation and coordination to characterize their results, but would be relevant to, beforehand, state what is the expected/efficient behaviour of individuals in this interaction.

3) it is hard to grasp the meaning of several expressions used in the text, namely “Uncertain thresholds” or “risk of thresholds”. The threshold seems to be defined on the stock size, below which fish stock growth rate is reduced. Events can occur and reduce fish stock growth rate below the threshold. It seems that there can be risk and uncertainty on the future growth rate, but the threshold is always defined at 28 or 20. How come the threshold is said to be uncertainty or risky?

4) one key conclusion of the paper is that “Fishers do reduce fishing in presence of thresholds”. But no scenario without thresholds was tested, as far as I understood. With or without event, the fish reproduction rate is always defined by thresholds defined on the stock size; lower growth rates when the stock approaches depletion is a feature present in all control and treatment scenarios. How can then be argued that “thresholds” increase cooperation? Compared with what?

5) seems rather strange that cooperation increases even in the baseline condition, where the second-stage is exactly the same as the first one. Actually, from the first to the second stage, the stock is replaced, which would imply that users are free to extract more without consequences. Any hypothesis for cooperation increasing also in the baseline condition?

6) if all treatments have threshold, why is a specific one called “threshold”? It seems that the distinctive feature of that treatment is for the event to occur deterministically.

Overall, I believe that this work can become a good contribution to PLOS ONE. In my opinion, it should be revised to clarify, at least, the points mentioned above.

Minor details:

Line 80, page 2: Fishers facing thresholds tend to fish less -> compared with what?

Page 3, line 114: cooperation is maximized when C = 1 -> seems that efficiency is maximized when C=1. As the authors mention, C<1 means cooperation as well.

Fig. 1 caption: When the difference were -> was?

Fig. 1 caption: contorl

Line 130, page 3: e regressed variables that summarizes -> summarize?

Line 180, page 5: Our findings supports -> support?

Page 5: wihtout crossing

Fig. 3 caption: starndard

SI: Page 1 line 26 -> sing up

Reviewer #2: This manuscript presents experimental evidence on the cooperative behaviours of individuals when faced to the risk of environmental crisis. Its main contribution is to provide insights on the behaviours of individuals used to manage resources, in contrast to previous work mostly focusing on individuals from WEIRD societies.

On the positive side, the manuscript provides a valuable contribution because (i) experimental evidence is always relevant (and this is particularly true in the field of evolution of cooperation dominated by theoretical work), and (ii) because the results of this study challenge previous conclusions mostly based on WEIRD societies. The introduction and the discussion are well written and referenced. The goal of the manuscript is well motivated. Previous work on the topic are cited and the authors do a good work at contextualising their study, either to motivate the study as in the introduction, or to connect their results in the discussion. The experimental design is solid. The authors show an expertise in statistics and data analysis, even if the complexity of the statistical analysis can sometimes limit the understanding of a reader.

The negative point is mainly the results section and the analysis (besides the statistical analysis) presented. First, some choices in the experimental design and the analysis lack justification (or alternatively, discussion of the consequences of these choices). For instance, the index created to measure cooperation appears arbitrary (arbitrary value in some cases, replenishment rate not taken in account, …). The rationale behind the effects of the climate event is not clear. Authors sometimes use median and sometimes use the mean. Altogether, this can lead readers to doubt the robustness of the conclusion of the study. Second, the results section strongly needs rewriting. On the one hand, a large part of the results section is not result but the description of the method. On the other hand, the space dedicated to the actual results and analysis is limited with for instance, the main result on cooperation appears to be missing. Moreover, the plots are too complex and often not clear. Ultimately, this results in readers having to trust the conclusions and interpretations of the authors rather than reaching the same conclusions than the authors throughout the analysis.

To conclude, I favour publication, but at the condition of rewriting the results section. The detailed comments are in the attachment.

6. PLOS authors have the option to publish the peer review history of their article (what does this mean?). If published, this will include your full peer review and any attached files.

Reviewer #1: No

Reviewer #2: **Yes: **Cedric Perret

---

## [Author Response · Author response to Decision Letter 0]

16 Oct 2020

## Detailed responses to editorial requests:

We have followed the suggested templates

Added in S3 appendix

All data necessary for replication has been updated under the Figshare repository 10.6084/m9.figshare.12563549. The anonymized data includes all the decisions taken in the experimental game (rawdata, N=4096), and a reduced data from our surve (mini_survey, N=256) that includes all questions used in our linear regressions. We did not include the full survey because we did not use all questions in the analysis, and some questions reveal delicate information about e.g. being victims of displacement or violence by illegal armed groups. 

5. Please amend your list of authors on the manuscript to ensure that each author is linked to an affiliation. Authors’ affiliations should reflect the institution where the work was done (if authors moved subsequently, you can also list the new affiliation stating “current affiliation:….” as necessary).

All affiliations listed are current.

6. We noted in your submission details that a portion of your manuscript may have been presented or published elsewhere. [We are also submitting another manuscript based on the same data entitled “Uncertainty can help protect the local commons in the face of climate change” also available as a preprint at https://papers.ssrn.com/sol3/papers.cfm?abstract_id=3468677. ] Please clarify whether this [conference proceeding or publication] was peer-reviewed and formally published. If this work was previously peer-reviewed and published, in the cover letter please provide the reason that this work does not constitute dual publication and should be included in the current manuscript.

The manuscript “Uncertainty can help protect the local commons in the face of climate change” is part of Caroline Schill’s PhD thesis defended at Stockholm University in 2016. The manuscript, from its initial draft, has benefited from the feedback from her PhD review committee, and was presented at the 6th World Congress of Environmental and Resource Economists in Gothenburg in 2018, and the International Conference for the Study of the Commons in Peru in 2019.

While relying on the same experiments, both papers differ in terms of the subset of data and methods (statistical analyses) used, level of analysis, metrics used, and scope of the conclusions. The paper led by Schill deals with how groups exploited shared resources in situations where climate change can introduce tipping points. It is a group level analysis, thus it focuses on overall sustained stock sizes as well as whether or not groups risk crossing the threshold as main variables of interests. Moreover, it uses inequality in catch rates (Gini coefficient) as a proxy for group cooperation. Schill’s paper has the advantage that group behaviour is independent among groups, lending itself for a different statistical approach (logit regressions with group level statistics). The disadvantage is that we cannot include individual level statistics such as the survey data, or individual risk and ambiguity preferences that we use on our PlosONE contribution.

Our paper submitted to PlosONE complements this group-level analysis used in the other manuscript by using individual-level information from post-experimental surveys, and risk / ambiguity elicitation tasks to better understand individual human behaviour in our experimental setting. Due to our main interest in the effects of thresholds on cooperation, we developed and introduced here a sophisticated metric of cooperation (different from Schill’s paper), which allowed us to disentangle cooperation from coordination effects at the individual level. For example, the PlosONE paper enables the distinction between cooperators and defectors that is fundamental in the cooperation literature, but inaccessible to the group level analysis. The PlosONE paper also develops a different methodological contribution to deal with dependencies in time, groups and individuals (3 potential sources of bias) on dynamic games. The manuscript led by Schill has not been peer-reviewed and published yet. We are currently in the process of getting it ready for re-submission. We will inform you if this status changes in the course of the revision process for this paper. 

## Detailed response to reviewers:

### Reviewer 1:

The authors present and discuss empirical results of an experiment with fisherman in Columbia, conducted to study individuals’ decision in a common-pool resource dilemma. Particularly, the authors design treatments to better understand how cooperation depends on shocks in the resource growth, thresholds that determine that resource growth, uncertainty and risk. The authors show that, in all the conditions tested, fisherman cooperate more after the resource growth regime shifts. Notwithstanding, the risk condition is the one presenting a higher increase in cooperation. The authors conclude that uncertain thresholds increase the level of cooperation.

This work has several positive aspects: 1) first of all, it is noteworthy that the common-pool resource game is actually played by individuals that deal with such dilemmas in their daily lives; this makes, in my opinion, the collected data extremely valuable. 2) second, it is also remarkable that the experiments were conducted with “non-weird” subjects. 3) last but not least, the results are interesting in that they point out that uncertain thresholds increase cooperation. The analysis performed appears to be sound and the results are, as far as I know, original.

Despite several worthy features, I also found some unclear points in the manuscript:

1) to start with, the authors use a very specific growth structure, characterized by (not one, but) four thresholds defining different fish stock growth rates. What was the reason behind such specific thresholds and growth rates?

We should point out that our game builds on a dynamic common-pool resource game originally designed for the lab (i.e. with students as participants) by Lindahl et al. (2016), and further developed by Schill et al. (2015). The point of reference for the underlying growth function of that design is a logistic growth function with a sigmoid term (in particular a “Holling-type” III predation term to capture resource dynamics with a threshold. Such a non-concave growth function has been shown to approximate the dynamics of ecosystems with the potential for regime shifts, such as forests or coral reefs. Like Lindahl and Schill et al. we use for our study also a discrete version of this growth function in order to reduce complexity of experiment instructions.

According to our definition of thresholds, our study has only one threshold. Below a stock size of 28 fish, the fish resource recovery rate drops from 10 to 1 fish. Here the use of “threshold” is understood as a critical point below which the dynamics of the system change to a qualitatively different behaviour (a bifurcation or critical transition). The other discrete steps in the recovery rate correspond to natural dynamics of ecological populations, including fish. Low recovery rate at higher densities ensures that the population has a carrying capacity, that is, it cannot grow to infinity. It reflects the situation when too many fish are already in the system and available resources become scarce for them. Low recovery rate at lower densities reflects depensation, or the fact that it is difficult for individuals to find partners for mating, and crowding benefits such as schooling in fishes are lost when the population has low numbers. As a result, fishes spend more time hiding or avoiding predators than feeding and reproducing, known in the literature as the Allee effect (Allee 1932). These two assumptions are necessary to keep the realism of our experiment: the population does not grow to infinity, and at lower densities it is harder to reproduce. These two assumptions are also necessary conditions to recreate the maximum sustainable yield in our experimental design, that is, an intermediate population size at which the recovery rate is maximised (in our case from 20-28 stock size).

In our experimental design the possibility of a threshold is induced and framed as a climate event in three out of 4 treatments. Crossing the threshold however is a response to fishing effort by the group of players, not to the climate event alone. We explained the threshold and the different growth rates to our fisher participants through the game instructions. We further checked that fishers were familiar with regime shifts like dynamics (bifurcations) in the survey when asking whether they have experienced abrupt collapses in their resources. Most of them confirmed experience of regime shift dynamics. The theory of critical transitions requires that multiple equilibria can co-exist under similar conditions (parameters). These equilibria in our game are the high and low reproduction rates, and the similar conditions are the climate event.

Allee WC, Bowen E (1932). "Studies in animal aggregations: mass protection against colloidal silver among goldfishes". Journal of Experimental Zoology. 61 (2): 185–207. doi:10.1002/jez.1400610202.

2) related with the previous point: what are the general characteristics of the interaction that results from such thresholds, growth rates and number of players per group? (e.g., what would be a pareto optimal set of strategies? or fair? or Nash equilibria?). The authors provide metrics of cooperation and coordination to characterize their results, but would be relevant to, beforehand, state what is the expected/efficient behaviour of individuals in this interaction

The Pareto optima in our game are all sets of decisions that maintain the stock size at 28 (just before crossing the threshold) for treatments, and at 20 in the baseline. Given that players can only extract a discrete number of resources, the strategy often evolves towards a rotation scheme where the 10 fish reproduced per round are splitted equally among the 4 players over the long term. The Pareto optima then assumes a fair distribution of resources, an assumption that permeates to our metrics of cooperation. The Nash equilibria, on the other hand, depends on a number of assumptions that are violated in our experimental design. A Nash equilibrium exists in non-cooperative games when people cannot form coalitions. Our game allows communication and coalition forming. The dynamic feature of our game (that current decisions can affect the pay-off table of future decisions) make our game very sensitive to the last round effect. That is, an optimal (Nash) strategy is to collapse the resource if one believes the game is about to end. But since the end of the game is unknown to players, the Nash equilibria become taking all the fish stock on the first round to avoid others doing it before yourself. We intentionally designed the game to avoid the last round effect. We have discussed the relevance of these equilibria in previous theoretical work that introduces our game design. For the intuition underlying these expectations please consult Lindahl et al. (2016) and Schill et al. (2015) on which the design of our game is built. We have not included analytical clarifications on these equilibria because 1) they are described elsewhere, and 2) they do not clarify the key findings of our paper. 

3) it is hard to grasp the meaning of several expressions used in the text, namely “Uncertain thresholds” or “risk of thresholds”. The threshold seems to be defined on the stock size, below which fish stock growth rate is reduced. Events can occur and reduce fish stock growth rate below the threshold. It seems that there can be risk and uncertainty on the future growth rate, but the threshold is always defined at 28 or 20. How come the threshold is said to be uncertainty or risky?

The threshold is indeed the point at which the dynamics tips and if it exists (depending on the treatment) it is always at the same stock size, known by participants. Below a stock size of 28, the growth rate drops drastically (from 10 to 1 fish only). According to our definition (see our answer above in 1), there is no threshold at a stock size of 20.“Uncertainty” and “Risk” refers to the probability of a climate event occurring which would induce a threshold in the resource dynamics. In other words, uncertainty and risk of a threshold do not refer to where the threshold is located but whether or not a threshold exists. In the risk treatment, the probability that the conditions enabling the threshold (the arrival of a climate event) were 50-50 (p = 0.5), and that probability was constant and known to all. In the uncertainty treatment, the probability range is known (p = 0.1-0.9) but the exact p is unknown to all. In the risk and uncertainty treatment, at any time after round 6, the players do not know if the climate event has happened and, hence, whether the threshold is activated or not — unless they cross it and realise that the recovery rate is not as high as it used to be.That’s why we refer to “uncertain thresholds” or the “risk of thresholds”. The third treatment was “Threshold” or when the threshold situation arrived for sure, with p = 1.

We realised that some formulations in the text (e.g. where we explain our cooperation index) could indeed let the reader assume that also baseline had a threshold (at stock size 20). We corrected for this, see line 206 

4) one key conclusion of the paper is that “Fishers do reduce fishing in presence of thresholds”. But no scenario without thresholds was tested, as far as I understood. With or without event, the fish reproduction rate is always defined by thresholds defined on the stock size; lower growth rates when the stock approaches depletion is a feature present in all control and treatment scenarios. How can then be argued that “thresholds” increase cooperation? Compared with what?

We did test a scenario without a threshold. It is called baseline. Only the threshold, risk and uncertainty treatments have thresholds below which the growth rate reduces drastically. To highlight this clearly and early on in the paper, we revised the description of the treatments in the “Fishing game” section accordingly. See line 100-107. It is compared to the baseline treatment, where there is no threshold modifying the maximum sustainable yield (MSY) area of the recovery rate. As clarified in the previous question, the other levels in the recovery rate as function of stock size are simply the necessary assumptions to make the game biologically realistic. In fisheries, the MSY is defined as a concave continuous function between the recovery rate and the population size. Here to reduce the level of details when explaining the game, we simplified the concave function to a step function with 3 levels: 0 in the extremes, 5 in intermediate levels, and 10 in the MSY area. The simplification was made to make the game easier to understand and follow for the fishers.

5) seems rather strange that cooperation increases even in the baseline condition, where the second-stage is exactly the same as the first one. Actually, from the first to the second stage, the stock is replaced, which would imply that users are free to extract more without consequences. Any hypothesis for cooperation increasing also in the baseline condition?

Indeed, as time passes in the game, the stock size is reduced and even in the baseline scenario, it is not optimal to bring the stock size below 20 fishes. So people adjust their strategies to maintain the stock in the MYS area and thus maximise their own utility. That is why we observe fishing reduction in the baseline treatment. That is also why we used a difference-in-difference regression approach to detect treatment effects. Given that there is a decline in fishing effort in the baseline treatment, the real effect is whether the decline in the treatment is larger and significantly different from one expects to occur in the baseline. That’s indeed what we found.

6) if all treatments have threshold, why is a specific one called “threshold”? It seems that the distinctive feature of that treatment is for the event to occur deterministically.

We hope we could clarify well enough with our answers above that according to our threshold definition a discrete step in the underlying growth function is a necessary but not a sufficient condition for a threshold. As explained in our answer to comment 1) above, we use a discrete version of a logistic growth function with a sigmoid term to simplify the explanation of the game. A threshold, as we understand it, would be a discontinuous jump in a continuous function. That is, under certain parameter values in the dynamics of the system, two or more regimes can co-exist. Once the system hits that tipping point, the dynamics will jump from two (or more in certain cases) qualitative modes of behaviour, or equilibria. In our design, that qualitative change (or bifurcation) is reflected by changing the recovery rate from 10 to 1 below a stock size of 28. While the critical point is deterministic, its occurrence is stochastic depending on the treatment, because it can only be crossed if the climate event conditions are activated. That activation depends on a lottery that was played in every round in the second part of the game. We have added a figure with the recovery rates (as suggested by reviewer 2) to clarify the differences in recovery rates between treatments.

Overall, I believe that this work can become a good contribution to PLOS ONE. In my opinion, it should be revised to clarify, at least, the points mentioned above.

Minor details:

Line 80, page 2: Fishers facing thresholds tend to fish less -> compared with what?

Added text: “compared to the baseline”

Page 3, line 114: cooperation is maximized when C = 1 -> seems that efficiency is maximized when C=1. As the authors mention, C<1 means cooperation as well.

Indeed, it is the most efficient cooperative outcome. However since we are describing C (cooperation) in this sentence, we refer to 1 as being the maximum value possible that C can get while still being cooperation. Values > 1 means less cooperation under our formulation.

Fig. 1 caption: When the difference were -> was?

Change made to “was”

Fig. 1 caption: contorl

Corrected

Line 130, page 3: e regressed variables that summarizes -> summarize?

corrected

Line 180, page 5: Our findings supports -> support?

Corrected

Page 5: wihtout crossing

corrected

Fig. 3 caption: starndard

Corrected

SI: Page 1 line 26 -> sing up

Corrected

### Reviewer 2:

Review: Cooperation in the face of thresholds, risk, and uncertainty 

This manuscript presents experimental evidence on the cooperative behaviours of individuals when faced to the risk of environmental crisis. Its main contribution is to provide insights on the behaviours of individuals used to manage resources, in contrast to previous work mostly focusing on WEIRD societies. 

On the positive side, the manuscript provides a valuable contribution because (i) experimental evidence is always relevant (and this is particularly true in the field of evolution of cooperation dominated by theoretical work), and (ii) because the results of this study challenge previous conclusions mostly based on WEIRD societies. The introduction and the discussion are well written and referenced. The goal of the manuscript is well motivated. Previous work on the topic are cited and the authors do a good work at contextualising their study, either to motivate the study as in the introduction, or to connect their results in the discussion. The experimental design is solid. The authors show an expertise in statistics and data analysis, even if the complexity of the statistical analysis can sometimes limit the understanding of a reader. 

The negative point is mainly the results section and the analysis (besides the statistical analysis) presented. First, some choices in the experimental design and the analysis lack justification (or alternatively, discussion of the consequences of these choices). For instance, the index created to measure cooperation appears arbitrary (arbitrary value in some cases, replenishment rate not taken in account, ...). The rationale behind the effects of the climate event is not clear. Authors sometimes use median and sometimes use the mean. Altogether, this can lead readers to doubt the robustness of the conclusion of the study. Second, the results section strongly needs rewriting. On the one hand, a large part of the results section is not result but the description of the method. On the other hand, the space dedicated to the actual results and analysis is limited with for instance, the main result on cooperation appears to be missing. Moreover, the plots are too complex and often not clear. Ultimately, this results in readers having to trust the conclusions and interpretations of the authors rather than reaching the same conclusions than the authors throughout the analysis. 

To conclude, I favour publication, but at the condition of rewriting the results section. You can find below the detailed list of comments:

Title 

Change the title to make it clear that the paper presents experimental evidence. For instance, add “Experimental evidence on [...]” or “[...] in fishers communities from Colombia”. 

Title changed to: “Cooperation in the face of thresholds, risk, and uncertainty: experimental evidence in fisher communities from Colombia.”

Introduction 

Some details at the end of the introduction should be moved to the next section. Authors should reconsider having a method section between introduction and results rather than at the end. 

Thanks for the suggestion. We have rewritten the manuscript with a methods section after introduction and an extended results section.

The authors might be interested by theoretical work by Francisco C. Santos on the topic of cooperation with risk (for instance, https://www.pnas.org/content/pnas/108/26/10421.full.pdf) .

Thanks for the suggestion, we have added reference to the fascinating work of Santos and Pacheco. Thanks for the lead, we were not aware of their theoretical model that predicted some of our results in the public goods context, very relevant indeed!

Method 

A figure that explains the replenishment rate would be helpful. For instance, the figure could be a line representing the population size of the fish stock, divided in sections for the different replenishment rate.

We have introduced the replenishment rate in text and a new Figure 1 as suggested 

What are the effects of the diminishing returns? It could be argued that individuals taking a lot of fishes are actually cooperating because they reduce the population size down to the maximal productivity. If it does not matter e.g. it rarely happens, add a sentence stating it.

That case is included in our measure of cooperation under the assumption of equal sharing (because we divide by number of players in the group). So taking a lot of fish at the beginning, as long as it is not by taking advantage of others and aggregated extraction does not lead to crossing the threshold is considered cooperation. Cooperation (C) will have a value of 1 or less.

The climate event (i) reduces replenishment rate of low population size and (ii) reduces the interval of population size where the replenishment rate is optimal but does not affect the replenishment rate of this interval. What is the rationale behind this choice? For instance, why did not the authors consider that climate event simply reduces replenishment rate for any population size? As far as I know, this differs from most of theoretical work so how does this choice affects the results presented, and the comparison with theoretical work?

The main purpose of our study was to test how individual resource users behave in situations pervaded by thresholds when facing collective action dilemmas. In particular, we were interested to what extent fishers reduce their fishing effort in order to not contribute to cross the (potential) threshold. No matter whether the fishers face the baseline (no threshold) or one of the treatment conditions (potential threshold), it is best for the group to maintain stock sizes where the replenishment rate is the highest. Changing the recovery rate for larger population sizes would have created a confounding factor and reduced our ability to answer our research question. If we introduce both, a threshold and a lower recovery rate for high population sizes, we would have not been able to distinguish whether a change in response/behaviour was because of the threshold in population size (x-axis) or the change in recovery rate (y-axis).

The reviewer’s suggestion is excellent, and probably a good way forward to advance our understanding of the role of recovery rates in fishing behaviour. In the scope of our study, however, it would have created the necessity of including 3 treatments (with and without change on the y axis), and increase our sample size; making the test unfeasible with our limited budget.

An additional argument not to change homogeneously the climate effect to all population sizes, is that the effect of climate change as reported in the literature is not homogeneous. Changes in temperature affect species differently, and the effects can differ even within the same species depending on their lifestage. While there is general agreement that climate change and fishing pressure together have negative effects on marine food webs (Kirby et al 2009), it is still an open question whether climate alone is necessarily negative for food webs productivity (Buchholz 2019). For example, in Arctic food webs some authors predict an increase of primary productivity under climate change scenarios (Lewis 2020, Buchholz 2019), while others predict the opposite (Whitt et al 2020). For tropical ecosystems it is likely to be negative, but the response on the reproductive rate is species specific: some species will benefit from warming conditions while others will see their niche reduced. That differential response is what seems to happen in the sardine / anchovy shifts along the Peruvian and Californian coasts, where the shift in species abundances is driven by climate (ENSO oscillations) (Sugihara et al 2012).

Kirby, Richard R, Gregory Beaugrand, and John A Lindley. 2009. “Synergistic Effects of Climate and Fishing in a Marine Ecosystem.” Ecosystems 12 (4). SPRINGER: 548–61. doi:10.1007/s10021-009-9241-9.

Buchholz, Andrea Bryndum, Derek P Tittensor, Julia L Blanchard, William W L Cheung, Marta Coll, Eric D Galbraith, Simon Jennings, Olivier Maury, and Heike K Lotze. 2019. “Twenty‐First‐Century Climate Change Impacts on Marine Animal Biomass and Ecosystem Structure Across Ocean Basins.” Global Change Biology 25 (2). John Wiley & Sons, Ltd (10.1111): 459–72. doi:10.1111/gcb.14512.

Lewis, K M, G L van Dijken, and K R Arrigo. 2020. “Changes in Phytoplankton Concentration Now Drive Increased Arctic Ocean Primary Production.” Science 369 (6500): 198–202. doi:10.1126/science.aay8380.

Whitt, Daniel B, and Malte F Jansen. 2020. “Slower Nutrient Stream Suppresses Subarctic Atlantic Ocean Biological Productivity in Global Warming.” Proceedings of the National Academy of Sciences of the United States of America 117 (27): 15504–10. doi:10.1073/pnas.2000851117.

Lotze, Heike K, Derek P Tittensor, Andrea Bryndum-Buchholz, Tyler D Eddy, William W L Cheung, Eric D Galbraith, Manuel Barange, et al. 2019. “Global Ensemble Projections Reveal Trophic Amplification of Ocean Biomass Declines with Climate Change.” Proceedings of the National Academy of Sciences of the United States of America 116 (26): 12907–12. doi:10.1073/pnas.1900194116.

Sugihara, G, R May, H Ye, C h Hsieh, E Deyle, M Fogarty, and S Munch. 2012. “Detecting Causality in Complex Ecosystems.” Science 338 (6106). American Association for the Advancement of Science: 496–500. doi:10.1126/science.1227079.

Why does the fish stock is restored at turn 7? Did fishers know about this? Can it affect the results?

At the beginning of the game, we informed all participants that the game lasts several rounds and that it has two stages. We also told them that we will tell them when the first stage finishes and explain then what will happen in the second stage. In other words, they did not know beforehand that we would reset the stock size, i.e. results were not affected. After 6 rounds, we informed the fishers that they are now going to play the second stage of the game. Apart from the new information they received depending on which treatment their group was randomly allocated to, they were all told that we will reset the stock to 50 fish. The reason for restoring the stock is twofold. On the one hand it allows to make within-group comparisons (i.e. compare behaviour of the same individuals and groups before and after the treatments were introduced). Additionally, due to the dynamic nature of our game individual groups are likely to maintain different stock sizes over time and so we would have been faced with introducing treatments while some groups maintain very high stock sizes, others very low ones (e.g. below the not yet introduced thresholds). Hence, we could not disentangle to what extent a change in behaviour would be due to the introduction of a treatment or due to where exactly the group was at after 6 rounds. 

L266 “There was no reproduction ...” -> Move this sentence up, where you explain the different replenishment rates.

Sentence moved

L259: “The event was meant to reduce ...” -> “The event reduces ...”

Changed

L297: remove the “than expected” and “initially planned”.

deleted

Did the fishers know the details about the different replenishment rates?

The fishers had complete knowledge about the different replenishment rates as well as every other aspect of the game design. See the instructions protocol in the SM. We also did 2-3 practice rounds before the game to make sure they understood how the replenishment rate worked.

L299: Add a reference for similar surveys in the literature to motivate the choice of the

survey (if it exists).

It is very common to complement behavioural economic experiments with post-experimental questionnaires or surveys see e.g. Anderies et al. 2011. The survey we used in this project is partly based on and inspired by post-experimental surveys used by the author team in previous studies. In particular: Maldonado and Moreno-Sanchez (2016) and Schill et al. (2015). We added this motivation and references to the text in the survey section. 

In revising the paper, we also realised that it would be more useful for the reader to highlight in the survey section only the questions we actually use for this study rather than providing a detailed overview of all the sections of our large survey. We hope you agree with this revision.

References:

Anderies, J. M., M. A. Janssen, F. Bousquet, J.-C. Cardenas, D. Castillo, M.-C. Lopez, R. Tobias, B. Vollan, and A. Wutich. 2011. The challenge of understanding decisions in experimental studies of common pool resource governance. Ecological Economics 70(9):1571–1579.

Maldonado, J. H., and R. Del Pilar Moreno-Sanchez. 2016. Exacerbating the tragedy of the commons: Private inefficient outcomes and peer effect in experimental games with fishing communities. PLoS ONE 11(2):1–17.

Schill, C., T. Lindahl, and A.-S. Crépin. 2015. Collective action and the risk of ecosystem regime shifts: insights from a laboratory experiment. Ecology and Society 20(1):48.

Analysis

Why does the authors use median cooperation but mean extraction?

Because cooperation is not normally distributed and the center of its distribution has a special meaning in the context of our study . For example, the median of cooperation for the threshold treatment aligns almost perfectly with C = 1, an important reference point in our study (where cooperation is at its maximum). Hence for the purpose of our research question, magnitude of the effects are best described by the median rather than the mean. The mean extraction is not normally distributed either. But its distribution does not have an additional meaning in the context of our research question.

Check if this notation is “p = 0.1:0.9” is commonly used. Change to 0.1 < p < 0.9 instead?

We changed to the second choice as suggested.

L88 – L95. This should be in the method section.

We don’t understand this suggestion. L88-95 presents our first sets of results — that fishing effort is reduced under threshold, risk and uncertainty treatments — It shows that the result is consistent with different choices of correcting for robust standard error estimations. It also interprets the result in the light of the literature introduced earlier and cautious the reader about the confounding factors that the diff-in-diff regression does not address, then motivating the next set of results.

L99 -L128: This should be in the method section

We have moved the text to the methods section as suggested.

L99-105: There is no need to discuss the different definitions of cooperation in the literature. One sentence explaining cooperation and how it is defined in this study is enough.

See comment below:

The measure of cooperation used lacks justification. This can lead readers to doubt of the results on cooperation, which is problematic because this is the main result. I understand the difficulty to create an index that take in account both the number of fish taken and the situation of the fish stock (above or below the threshold). I would advise to either split the index in two, with one index describing if individuals take fish when the stock is below the threshold, and one index describing the amount of fishes taken when above the threshold. Alternatively, the author needs to better justify the index built.

In the broader literature, cooperation is often confounded as i) maintaining the resource, versus ii) following agreements. For example, in the classical one-shot prisoner's dilemma cooperation is taken as following an implicit agreement of non-defection. But in our game that definition is limiting because, as you point out, there are multiple dimensions to what cooperation means in a dynamic setting: when the choices of the present changes dynamically the payoff functions of the future. In economics, there is a handful of papers using dynamic games in common pool resource contexts; and most of them operationalise cooperation at the group level (not crossing the threshold) not at the individual one. 

Calling for a more evolutionary perspective in our definition of cooperation allows us to include the aspect of fitness (or fitness loss) in the long term, and control for the scenario when people agree to collapse the resource. Would that be cooperation? For some economists it is because agents are maximising their returns. But within our framework it is not because by reducing the ability of the stock to recover, a person is reducing her own profits in the future (reducing fitness) — regardless of what the group do. So our way of measuring it allows us to speak of cooperators at the individual level, distinguish from defectors, and contrast cooperation versus mere coordination.

The option suggested is not feasible at the individual level because the fish taken below the threshold is an attribute of the group, not the individual. For example if the fish stock is 30 and all 4 players take 2 fishes (2*4 = 8) who should be made responsible for taking the extra fish? Crossing the threshold is a group level feature, not an individual one. It also changes the statistical analysis where the group is the unit of analysis and the number of rounds above or below threshold is a response variable. That is what we did on a separate paper for the group level analysis by means of a logistic regression [pre-print available here: https://www.ssrn.com/abstract=3468677]. That analysis, however, cannot make use of the individual level information we present in our manuscript e.g. the surveys.

Your questions, however, raises the issue that our measure and what motivates it does not come sufficiently clear in the current text. We have added some clarifications following your suggestions below as well as made some additional changes that should improve our justifications.

o Provide the rationale behind the choice made to build the index: 

 ▪ For instance, “We consider that cooperation is represented by individuals

maintaining the fish stock in its most productive/sustainable state. Thus, cooperation happens when (i) individuals take a number of fishes that maintain the fish stock above the threshold, or (ii) take no fishes when the fish stock is below the threshold”

We adapted the text following your suggestion (see line XXX):

“Here we operationalise these definitions by measuring cooperation as the ratio of the individual extraction $x_{i,t}$ with respect to the optimal level for the group. Thus, cooperation happens when (i) individuals take a number of fishes that maintain the fish stock at the optimal level (i.e. above the threshold in the treatments), or when (ii) take no fishes when the fish stock is below the optimal level (i.e. below the threshold in the treatments)..”

 ▪ Using biological terms rather than mathematical terms helps readers understand the choice made to built the index. For instance: “To avoid division by zero or negative values, if the denominator is < 1 and x_i,t = 0 cooperation is set to C = 1 (212/4096 observations), [...]” can be replaced by “When the fish stock is below the threshold (denominator < 1) and fishers do not take any fish (x_i,t = 0), we consider that they cooperate C = 1”. 

Change made

o The authors consider that fishers taking fish while the stock is below the threshold results in a cooperation value of 1.5. This value is arbitrary so what does happen when a different value is used? 

This was an error in the text, the text should read “if the denominator is zero and x_i,t = 1, C = 1.5”. We introduced this rule to avoid division by zero. This seemingly arbitrary choice of value does not affect results in any way because it concerns only 17 / 4096 observations, i.e. 0.004% of the data. The important decision to follow here was to choose a value between 1 (maximum cooperation) and 2 (non-cooperative behaviour that could lead to the group crossing the threshold in one of the threshold treatments). An obvious choice for a value between 1 and 2 is 1.5. The error is corrected.

o It is confusing that a higher value of the cooperation index means lower cooperation. Change the index to (1 – C_i,t) or change the name of the index. 

I don’t think a change of name or scale would do the trick because the underlying function has a concave shape with a maximum in 1. Transforming 1-C centres the distribution to zero, but the magnitude of the scores loses interpretability. C = 2 means people took twice of what was fair, C = 1 means they took 100% of what was fair, and C = 0.5 means they took 50% of what was fair favouring the common good over maximising individual earnings. That numeric interpretation is lost with negative and positive values (-1 is not a negative equivalent of +1).

We have also emphasised that singular points in time were not good indicators of cooperation, but rather the distribution across time. This is because rotation schemes could emerge when an individual in a round is allowed to take more than what is fair, if she allows others to do the same in the future. So it is the shape of the distribution (and its median because the observations are not normally distributed) rather than C_i, which is interpreted as cooperation in our study.

o Why does the replenishment rate is not taken in account in the cooperation index? 

It is taken into account indirectly in the sign of the denominator for stock size values around the threshold. If the denominator is positive the replenishment rate is high. When the threshold has been crossed and the replenishment rate jumps to a lower value, the denominator is negative. Low values of the replenishment rate at high stock sizes are not taken into consideration in the cooperation index because these are not situations of scarcity or risk.

L120-123. It is not clear why the authors justify the choice of describing coordination this way. If the authors use the average cooperation across different rounds, deviation from cooperation for a single round should not be a problem. I would remove this part. Simply introduce coordination in a way to measure coordination.

We did not fully understand the comment. Our measure of coordination is simply to measure coordination, it is a distance measure on the fish extraction decisions (x_i,t). It does not take into account cooperation. There is a weak negative correlation between the two (shown in SFig 1), and as we argue above they measure different things. People can coordinate to collapse the resource or to maintain it above the threshold. Both strategies will score high on coordination, but cooperation will be low in the first case while high on the second. We introduced both measures to be able to distinguish between cases.

L120-123 introduces the reader to the case when coordination is high but not 1 (a rotation scheme emerged making choices among players similar), and cooperation is high on average (close to 1) but sometimes higher than 1 because of the rotation agreement (one or two players take more than what is fair while keeping the stock as a group above the threshold). The game does not allow people to fish non-discrete numbers of fish forcing sometimes such rotation agreements.

L128: Present the results of the effects of treatments on cooperation and coordination before starting to explain these effects. So far, the results section seems to lack the main result (no effect of treatments on cooperation)

We adapted the results section according to your suggestion regarding cooperation. The second sentence in the results section now reads as follows: “We also find that contrary to theoretical expectations, cooperation does not break down.” The result regarding coordination we have to report later (with the second set of regressions) as we cannot perform the DID random effects panel model with the variable coordination (because the similarity score is calculated over the rounds 1-6 for the first phase and 7-16 for the second).

L141: Not clear and this choice looks arbitrary. Add justification or reference.

The justification is that our experimental design focuses on the impacts of tipping points in natural resource dynamics, so for us it is more important how it affects the livelihood of fishers rather than their income per se. Our survey provides info on income as well as the frequency of bad days (or days without income); and of course they are correlated. We’ve chosen the second because it is closely related to the exposure of fishers to regime shift dynamics. Income per se might be compensated by other economic activities or alternative sources of income such as pension or remittances. The justification is in L288-92.

L149. Split the sentence for clarity. First part is about all treatments having an effect and second part is about a single treatment having an effect.

Modification added.

L157: But do individuals that reach agreement are explained by socio-economic effects? 

Thanks for the suggestion, really good idea that we have not checked before. We tested it now and the only socio-economic factor that explains the proportion of agreements is unsurprisingly education. The p-value is 0.002, but the size of the effect is rather minimal 0.01. It means that for every additional year of education, the subject is likely to reach agreements in 0.01 rounds. The proportion of agreements is measured between 0-1 and corresponds to the number of rounds in which the group made agreements in the treatment stage (10 rounds max). Because the effect is so little, we don’t think it is worth including in the main text. If the editor and reviewer deem it necessary, we are however happy to include the respective regression table in the SM. Below a raw summary:

Call:

lm_robust(formula = prop_ag ~ Treatment + Place + education_yr + 

 BD_how_often + fishing_children + Risk + Amb, data = ind_coop %>% 

 filter(part == T) %>% ungroup(), clusters = group, se_type = "stata")

Standard error type: stata 

Coefficients:

 Estimate Std. Error t value Pr(>|t|) CI Lower CI Upper DF

(Intercept) 0.509674 0.127783 3.98859 0.0001781 0.254239 0.76511 62

TreatmentThreshold 0.109000 0.099416 1.09640 0.2771433 -0.089730 0.30773 62

TreatmentRisk 0.011077 0.123237 0.08988 0.9286704 -0.235271 0.25742 62

TreatmentUncertainty 0.104112 0.108432 0.96016 0.3407063 -0.112641 0.32086 62

PlaceB -0.180979 0.118071 -1.53280 0.1304129 -0.417000 0.05504 62

PlaceC 0.095775 0.087121 1.09934 0.2758723 -0.078377 0.26993 62

PlaceD -0.003950 0.120360 -0.03282 0.9739244 -0.244546 0.23665 62

education_yr 0.016855 0.007177 2.34840 0.0220580 0.002508 0.03120 62

BD_how_often -0.001341 0.012102 -0.11084 0.9120992 -0.025533 0.02285 62

fishing_children 0.025059 0.044248 0.56635 0.5732031 -0.063390 0.11351 62

Risk 0.004053 0.012261 0.33058 0.7420790 -0.020457 0.02856 62

Amb -0.010571 0.014910 -0.70897 0.4809979 -0.040377 0.01923 62

Multiple R-squared: 0.2235 , Adjusted R-squared: 0.1871 

F-statistic: 2.083 on 11 and 62 DF, p-value: 0.03487

The results section ends without having the main result (no effect of treatment on cooperation) being clearly presented. It appears that the result section starts by describing the effect of treatments on the number/proportion of fishes taken and then jumps directly to how these effects can be explained (socio-economic factors, coordination and agreement).

As explained above, to another of your comments, the second sentence in the results section now reads as follows: “We also find that contrary to theoretical expectations, cooperation does not break down..” Furthermore, we clearly repeat this result now at the beginning of the third paragraph of the results section: “Besides the effects of treatments on the reduction of fishing effort, we find that cooperation does not break down .” This aligns with the caption used in Fig 2 to describe the main results of our paper.

Figures

• Figure 1:

What is before and after? I do not find explanations in the text. Does that mean that the cooperation presented is averaged on the rounds before and after the round 7?

Indeed, before and after corresponds to the introduction of the treatment in round 7. We included that information in the caption of the figure. The technical explanation and formulas to calculate the diff-in-diff regression are introduced in the Regressions section under the Methods. We added the number of observations to the caption. 

I would advise to start the caption by a sentence presenting the plots, and then have a sentence describing more formally the analysis. For instance, “Effect of treatment (risk, threshold, uncertainty) on the individual extraction (top), proportion of stock (middle) and cooperation (bottom). The effects of the treatment are tested using ...” 

We appreciate the suggestion but given that our main message seemed lost in your previous comments, we prefer to keep the start of the caption as is. It encapsulates our main result: fishers fish less but cooperation does not change. However, we do see the benefit of clearly referring to the three treatments versus baseline in the caption. We adapted the caption accordingly. 

Replace “counterfactual” by “baseline” in the line type (or explain in the caption). 

The counterfactual is not the baseline per se, it is what people in the treatment would have done if they would have played the baseline instead of the treatment. It relies on the parallel assumption of the diff-in-diff identification strategy. The counterfactual is not actually observed, it is inferred (see table with formulas in the methods). For more details we recommend: 

Angrist and Pischke. 2009. Mostly harmless econometrics. Princeton Press

We have amended the caption with the clarification as suggested.

• Figure 2 is not clear at all. 

First, it can be improved in term of appearance, e.g. the quality is low, the number of different plots is too high, the size of the plots change. 

Second, a plot needs to support one or two conclusions rather than providing an exhaustive presentation of the results (this goes into supplementary materials). Split this figure in different figures. 

Thanks for your comments and suggestions. We improved and simplified the figure accordingly.

• Figure 3

o If possible, colour the points as a function of their p-values, in the same way than Figure 1. 

Suggestion implemented.

Discussion 

• L175: First sentence is not clear. 

L175: “Fishers under uncertain thresholds maintained higher levels of cooperation than when the risk of thresholds was known, but risk had a stronger effect at reducing individual fishing effort than uncertainty.” L207: “However, cooperation as measured in our study was not affected by our treatments.”. These two statements seem to contradict each other. 

Indeed, there was a mistake. We have rephrased the first sentence as follows: “Fishers under uncertain thresholds showed lower levels of extraction than when the threshold was known. Risk had a stronger effect at reducing individual fishing effort than uncertainty.”

The second part of the statement was left unchanged. 

L180: The authors state that uncertainty increase cooperation, but I thought that uncertainty did not affect the level of cooperation.

L180 reads: “Our findings supports the hypothesis that uncertainty can increase cooperative behaviour in public good settings when the value of the public good is sufficiently high”.

What we report is that cooperation does not decrease — does not break down. And it can increase, as suggested previously under certain circumstances, when it matters a lot to people e.g. their livelihood depends on it. We observe signals of increasing cooperation in the form of reduction of fishing effort as uncertainty increases. We observe both displacement of the distribution to C values 0-1 in Figure 2, and effects on reduction of variance in extraction (Figs 2 and 3). Cooperation increased but the differences to the counterfactual were not significant (Fig 1). Our main point is that cooperation does not break down. The studies on public goods measure cooperation differently (the size of contribution to the public good). Here they are compared with the size of reduction in use to the CPR. 

L211: The author could cite Elinor Ostrom, e.g. Governing the commons (1990).

Reference added 

Typo

A comma is often missing “,”: L316: “For risk the chances [...]”, L360: “As response variables we used […]”.

Commas added

Fig 1 caption: “contorl”. Corrected

L184 “effots”. Corrected 

Fig2 caption: Add “Figure” before “A) and B)”. Added

---

## [Decision Letter · Decision Letter 1]

2 Nov 2020

Cooperation in the face of thresholds, risk, and uncertainty: experimental evidence in fisher communities from Colombia

PONE-D-20-19709R1

Dear Dr. Rocha,

We’re pleased to inform you that your manuscript has been judged scientifically suitable for publication and will be formally accepted for publication once it meets all outstanding technical requirements.

Kind regards,

The Anh Han, Ph.D.

Academic Editor

PLOS ONE

Additional Editor Comments (optional):

Both reviewers are happy with the responses provided and changes made, by the authors. There are only some very minor aspects suggested by the reviewers but I bellieve they can be dealt with when preparing the final version of the manuscript.

Reviewers' comments:

Reviewer's Responses to Questions

**Comments to the Author**

1. If the authors have adequately addressed your comments raised in a previous round of review and you feel that this manuscript is now acceptable for publication, you may indicate that here to bypass the “Comments to the Author” section, enter your conflict of interest statement in the “Confidential to Editor” section, and submit your "Accept" recommendation.

Reviewer #1: All comments have been addressed

Reviewer #3: All comments have been addressed

2. Is the manuscript technically sound, and do the data support the conclusions?

Reviewer #1: (No Response)

Reviewer #3: Yes

3. Has the statistical analysis been performed appropriately and rigorously? 

Reviewer #1: (No Response)

Reviewer #3: Yes

4. Have the authors made all data underlying the findings in their manuscript fully available?

Reviewer #1: (No Response)

Reviewer #3: Yes

5. Is the manuscript presented in an intelligible fashion and written in standard English?

Reviewer #1: (No Response)

Reviewer #3: Yes

6. Review Comments to the Author

Reviewer #1: Thanks to the authors for the clarifying answers and the changes made in the manuscript. I am satisfied with the answers provided.

I found useful the explicit connection between the game studied and the previous works Schill (2015) [17] and Lindahl (2016) [21].

The answers provided clarify the distinction between the abrupt changes in reproduction rates that are due to 1) natural dynamics or that are due to 2) climate events. The authors only use “threshold” to refer to the latter. By reading the manuscript, however, I still found hard to apprehend the precise meaning of "threshold" in this work. Perhaps the new Figure 1 can be used to clarify this. The authors could mention in the caption that the discontinuities observed in the left panel only reflect natural reproduction dynamics. The caption of Fig. 1 in Schill (2015) [Ref 17] does a good job defining the meaning of threshold and some description along those lines could be added also here.

Reviewer #3: I would like to deeply thank the authors for their detailed answer. The authors have made the modifications suggested, and when they did not, they have well justified their choices or pointed out my mistakes or misunderstanding. I am in favour of publication.

For the comment on the role of socio-economic factors on the proportion of agreements, I let the authors take the decision on whether to include it in the manuscript. I thank the authors for taking the time to do this extra analysis. A last comment is that at line 87, there is a paragraph on the ethical review process. Ignore me if it is required to have this paragraph in the main text, but if not, I think the authors could move it to one of the supplementary materials.

7. PLOS authors have the option to publish the peer review history of their article (what does this mean?). If published, this will include your full peer review and any attached files.

Reviewer #1: No

Reviewer #3: **Yes: **Cedric Perret

---

## [Editor Report · Acceptance letter]

17 Nov 2020

PONE-D-20-19709R1 

Cooperation in the face of thresholds, risk, and uncertainty: experimental evidence in fisher communities from Colombia 

Dear Dr. Rocha:

I'm pleased to inform you that your manuscript has been deemed suitable for publication in PLOS ONE. Congratulations! Your manuscript is now with our production department. 

Kind regards, 

on behalf of

Dr. The Anh Han 

Academic Editor

PLOS ONE